# Revisiting Random Generation Order: Ordinal-Biased Random Training for Efficient Visual Autoregressive Models

## Abstract

We observe an interesting **Ordinal Asymmetry** phenomenon when training visual autoregressive (AR) generators with randomized generation paths: *early tokens*, due to limited context, suffer higher losses and primarily capture *global structure*, while *later tokens*, with richer context, incur lower losses and refine *local detail*. This suggests that conventional randomized training must optimize **two qualitatively different ordinal subproblems** at once . From a curriculum perspective, training can focus on one of the two ordinal subproblems instead of optimizing both simultaneously. Therefore, we propose **Ordinal-biased Random Training** (ORT), a simple strategy that first biases loss weights toward later tokens or early tokens and then gradually anneals to uniform weighting, ensuring that both global structure and local detail are learned. Specifically, we implement ORT with an *ordinal focal loss* that assigns position-dependent weights; the schedule is controllable and can emphasize either early or late tokens. In practice, ORT shows a striking *sudden convergence*: gradient norms collapse sharply during the middle of randomized-path training, providing clear evidence that late-biased weighting accelerates early-stage optimization. Experiments on ImageNet-256 with RAR validate our analysis: ORT halves the randomized training phase ($200 \rightarrow 100$ epochs, $2\times$ faster) while maintaining FID comparable to the 400-epoch RAR-XL baseline.

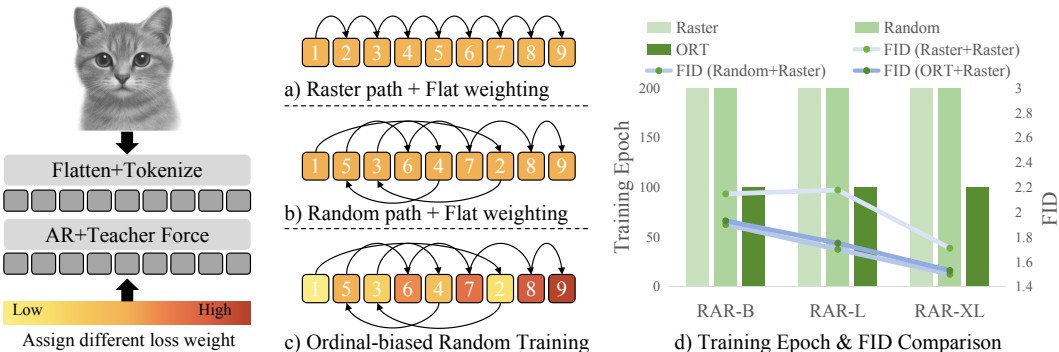

Figure 1: Comparison of Ordinal-Biased Training (ORT) with prior works. **(a)** Raster path enforces a fixed scan order. **(b)** RAR (Yu et al., 2024b) adopts random order but treats all positions equally. **(c)** $\text{ORT}_l$ (ours) re-weights token-level loss by ordinal position. $\text{ORT}_l$ favors later tokens, whereas $\text{ORT}_e$ emphasizes early tokens; both provide effective training speedups. **(d)** On ImageNet-256 with RAR, $\text{ORT}_l$ halves the randomized phase while maintaining comparable or better FID.

## 1 Introduction

Visual autoregressive generation models (Esser et al., 2021a; Luo et al., 2024; Sun et al., 2024; Li et al., 2024; Tian et al., 2024) decompose visual data into raster-order sequences of discrete tokens and train models to predict each token based on previously observed ones via teacher forcing (Williams & Zipser, 1989). To better reflect the inherently 2D structure of images and remove the raster-order spatial bias (left-to-right and top-to-bottom order), recent methods such as RAR (Yu et al., 2024b)

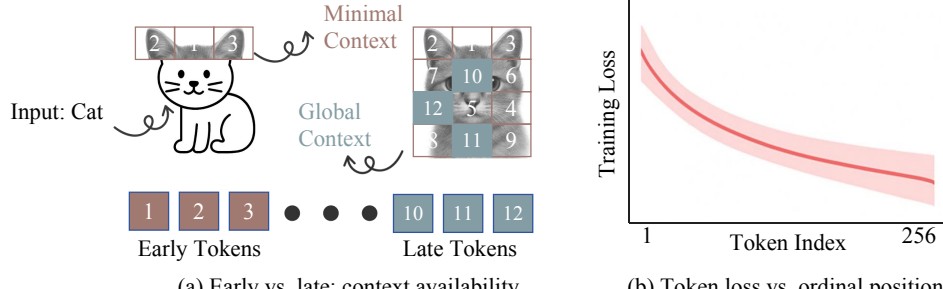

(a) Early vs. late: context availability.

(b) Token loss vs. ordinal position

Figure 2: Ordinal asymmetry under randomized-path training. **(a)** Early tokens are generated with minimal context, whereas later tokens benefit from accumulated global context. **(b)** Token-level training loss decreases with ordinal index, indicating that early tokens are harder to optimize compared to later tokens.

and RandAR (Pang et al., 2025) introduce randomized generation paths. These enable bi-directional context modeling of 2D images while preserving the autoregressive (AR) training paradigm.

Despite these improvements, existing approaches share a common assumption: *all token positions are treated equally during training*, with uniform loss weighting and supervision across the sequence. However, under randomized generation paths, the context available to each token varies significantly with its position in the sequence (*i.e.*, early tokens have little to no context, while later tokens can access almost the entire image, as illustrated in Fig. 2 (a)). This raises a fundamental question: do tokens at different positions play different functional roles during learning?

In this work, we revisit randomized generation orders for training visual autoregressive (AR) models and uncover a consistent pattern, referred to as **Ordinal Asymmetry**: token positions in the sequence are not equivalent in their optimization objective and difficulty. The predictions of early token positions, generated with little or no context, account for determining global structure, while that of later positions, benefiting from generated tokens, focus on local detail refinement (Fig. 2). To analyze their different roles, we conduct shortened training runs (50 epochs instead of the full 400) and vary loss weighting across ordinal positions, which is shown in Table 1. We observe a trade-off in the early training stage: emphasizing early tokens' losses improves global fidelity (FID ↓17%) but not local coherence (sFID ↓4%), whereas emphasizing later tokens' losses improves local quality (sFID ↓13%) at the cost of global structure (FID ↑13%). These findings indicate that in the early training phase, ordinal asymmetry manifests in both objective and difficulty: the predictions of early positions aim to establish global structure but are harder to optimize, whereas the predictions of later positions refine local details and are easier to learn.

The ordinal asymmetry above can be naturally interpreted through the lens of curriculum learning (Soviany et al., 2022), where models can focus on an easier subproblem (global or details) instead of optimizing both simultaneously. This perspective motivates a simple acceleration strategy: temporarily emphasize either early or late tokens to *warm up* training, then gradually expand supervision to all tokens once the model has stabilized. Building on this idea, we introduce **Ordinal-biased Random Training** (ORT), which instantiates this curriculum via an ordinal focal loss that linearly reweights tokens according to their generation positions. Despite its simplicity, this position-dependent scheme offers an effective way to exploit the inherent ordinal structure of AR generation. We validate ORT on ImageNet-256, where the original RAR schedule trains for 400 epochs with 200 epochs dedicated to randomized training. With late-biased weighting, the randomized phase is reduced from 200 to 100 epochs and we observe a pronounced *phase transition* in gradient norms around this point; after 300 epochs the model reaches FID 1.53, comparable to the ∼1.51 of the full 400-epoch RAR-XL baseline.

Our contributions are listed as follows:

**(1)** We revisit randomized generation orders and uncover an intrinsic *Ordinal Asymmetry*, showing that token positions differ fundamentally in their optimization dynamics.

**(2)** Inspired by a curriculum perspective, we propose **Ordinal-biased Random Training** (ORT), a simple reweighting strategy that emphasizes late tokens or early tokens only in the early stage.

**(3)** We conduct extensive experiments and analysis on ImageNet-256, which validate our analysis and show that the proposed approach not only improves training efficiency but also achieves competitive generation quality compared to the state-of-the-art AR and diffusion models.

## 2 RELATED WORK

**Autoregressive Modeling in NLP.** Autoregressive (AR) models are central to natural language processing (NLP), enabling significant advancements in language modeling. Early models like GPT-2 (Radford et al., 2019) and GPT-3 (Brown et al., 2020) showcased AR's strength in generating coherent and fluent text by producing tokens sequentially, each conditioned on prior tokens. This sequential approach effectively captures linguistic dependencies, making AR models well-suited for text generation tasks. Innovations like T5 (Raffel et al., 2020) have further expanded AR's applicability by reformulating various NLP tasks as text-to-text transformations, enhancing their adaptability across diverse applications.

**Autoregressive Modeling in Visual Generation.** Autoregressive (AR) models have long been used in visual generation, producing images sequentially by conditioning each token or pixel on those previously generated. Early work such as PixelCNN (Van den Oord et al., 2016) demonstrated that AR factorization can capture fine-grained spatial structure and yield high-quality samples. Recent progress has further advanced AR-based generation through improved visual tokenizers (Yu et al., 2024c) and redesigned generation paradigms (Tian et al., 2024; Pang et al., 2025; Yu et al., 2024b). VAR (Tian et al., 2024) reframes AR generation along a coarse-to-fine scale hierarchy rather than over token level. MAR (Li et al., 2024) adopts masked autoregression with a continuous diffusion loss to mitigate artifacts from discrete tokenization. RAR (Yu et al., 2024b) leverages randomized generation paths to recover bidirectional prior that traditional raster-order training fails to model.

**Image tokenizer.** Learned visual tokenizers (often implemented as variational or quantized autoencoders) have become central to modern generative visual models for both autoregressive (Yu et al., 2024b; Wu et al., 2025) and diffusion-based (AI, 2023) generator. These models condense raw pixel arrays into compact latent representations, enabling downstream generators to operate efficiently in a lower-dimensional space. Early discrete tokenizers such as VQ-VAE (van den Oord et al., 2018) introduced vector quantization to map continuous latents into code indices, allowing autoregressive decoders to model images in a discrete sequence space. Follow-up work, including VQ-GAN (Esser et al., 2021b), improved perceptual quality through adversarial training, while more recent approaches, such as FSQ (Mentzer et al., 2023) and MagViT-v2 (Yu et al., 2024a)) focus on scaling codebooks and stabilizing training at larger capacities. Parallel efforts explore alternative tokenization paradigms: TiTok (Yu et al., 2024c) replaces convolutional encoders with a Transformer-based patch tokenizer that outputs a single 1D token sequence, and FlexTok (Bachmann et al., 2025) investigates hierarchical tokenization that captures multi-scale semantics.

## 3 PRELIMINARY

Autoregressive (AR) models decompose the generation of visual data (images or videos) into a sequential prediction task. Given a rasterized token sequence $\mathbf{x} = \{x_t\}_{t=1}^{T}$ obtained by partitioning the image into $T$ discrete tokens (e.g., via VQ Tokenizer (Chang et al., 2022) in our baseline RAR (Yu et al., 2024b)), the joint distribution $\mathbf{x}$ factorizes as:

$$p_\theta(\mathbf{x}) = \prod_{t=1}^{T} p_\theta(x_t|x_{1:t-1}), \tag{1}$$

where $x_{1:t-1} = x_1, ..., x_{t-1}$ denotes previous tokens, $\theta$ is an AR model with vision transformer. The model $\theta$ predicts the distribution of the next token $x_t$ as:

$$p_\theta(x_t|x_{1:t-1}) = \text{softmax}(h_t), \tag{2}$$

where $h_t = \text{Transformer}(\mathbf{E}(x_{1:t-1}))$ and $\mathbf{E}(\cdot)$ denotes the token embedding function. The training objective minimizes the negative log-likelihood:

$$\mathcal{L}_{\text{AR}} = -\frac{1}{T}\sum_{t=1}^{T} \log p_\theta(x_t|x_{1:t-1}). \tag{3}$$

During training, the full token sequence is input simultaneously, and the autoregressive constraint is enforced via a causal attention mask that prevents each token from accessing future tokens.

## 4 REVISITING RANDOMIZED GENERATION ORDERS: ORDINAL ASYMMETRY

### 4.1 MOTIVATION AND OBSERVATIONS

In standard autoregressive models, tokens are generated in a fixed order (e.g., raster scan). Recent works (Yu et al., 2024b; Pang et al., 2025) introduce a randomized generation order to improve generation diversity and robustness. Formally, let $\pi = \{\pi_1, \pi_2, ..., \pi_T\}$ be a permutation of $\{1, 2, ..., T\}$, representing a randomly sampled generation path. The joint distribution is then factorized as: $p_\theta(\mathbf{x}) = \prod_{t=1}^{T} p_\theta(x_{\pi_t} \mid x_{\pi_{<t}})$, where $x_{\pi_{<t}} = \{x_{\pi_1}, ..., x_{\pi_{t-1}}\}$ denotes the set of previously generated tokens in the sampled order $\pi$. The corresponding training objective becomes:

$$\mathcal{L}_{\text{RandomAR}} = -\frac{1}{T} \sum_{t=1}^{T} \log p_\theta(x_{\pi_t} \mid x_{\pi_{<t}}). \quad (4)$$

Interestingly, we observe a consistent phenomenon: early tokens in the generation order tend to have significantly lower prediction probability (*i.e.*, $p_\theta(x_{\pi_t} \mid x_{\pi_{<t}})$ in Eq. 4) as shown in Fig. 3. This indicates that these early tokens are more uncertain and challenging for the model to predict. We hypothesize that early tokens are harder to optimize because they lack contextual cues and must account for global structure, whereas later tokens benefit from accumulated context and primarily handle local refinement.

To examine this hypothesis, we introduce Ordinal Focal Loss (ORT) that adaptively up or down-weights tokens based on their generation order. By reweighting the loss according to ordinal position, we test whether emphasizing specific subsets of tokens (e.g., early or late) can improve different aspects of generation quality, such as FID and sFID.

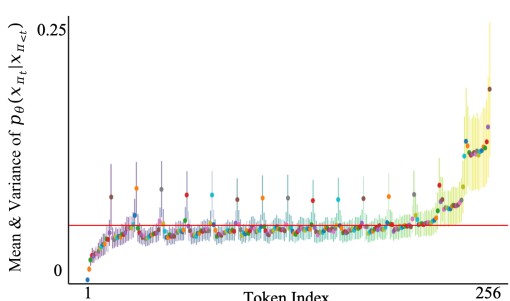

Figure 3: Mean and variance of the token probabilities $p_\theta(x_{\pi_t} \mid x_{\pi_{<t}})$ across different ordinal positions. Each colored dot shows the mean confidence for tokens generated at position $t$, with vertical bars indicating the standard deviation. The red horizontal line marks the global average confidence across all tokens. The pattern shows that earlier (left) and very late (right) positions exhibit distinct mean probabilities and variance, suggesting differing levels of prediction difficulty.

### 4.2 ORDINAL FOCAL LOSS

Motivated by the above findings, we introduce *Ordinal Focal Loss*, a generic weighting scheme to adaptively emphasize different ordinal positions. Instead of uniform averaging in Eq. 4, we reweight each token's loss according to its position $t$ in the generation path:

$$\mathcal{L}_{\text{OrdinalFocal}} = -\frac{1}{T} \sum_{t=1}^{T} \omega(t; \alpha, \beta) \cdot \log p_\theta(x_{\pi_t} | x_{\pi_{<t}}), \quad (5)$$

where the weight function $\omega(t; \alpha, \beta)$ linearly interpolates between $\alpha$ (for $t = 1$) and $\beta$ (for $t = T$):

$$\omega(t; \alpha, \beta) = \alpha + (\beta - \alpha) \cdot \frac{t-1}{T-1}. \quad (6)$$

This formulation enables the model to focus more on early (global) or late (local) tokens depending on the downstream needs, while remaining fully compatible with existing random-order AR frameworks.

### 4.3 IMPACT ON GENERATION QUALITY

To better understand the influence of ordinal weighting strategies, we conduct an extensive analysis by sweeping different values of the linear weighting endpoints $\alpha$ and $\beta$. Specifically, we evaluate a range of $(\alpha, \beta)$ pairs, each defining a linearly increasing or decreasing loss weight from the first token ($t = 1$) to the last token ($t = T$). This enables us to investigate how different emphases—either on early (global) or late (local) tokens—affect generation quality.

Table 1: Early-phase results and full (50 epochs, RAR-B (Yu et al., 2024b)on ImageNet-256 (Deng et al., 2009)) with different linear weighting strategies. The results indicate that tokens at different ordinal positions serve distinct roles: early tokens dominate global quality (FID), whereas later tokens contribute more to local quality (sFID, recall). Deeper colors represent better results.

| Focus on | $\alpha$ | $\beta$ | FID ↓ | IS ↑ | sFID ↓ | Precision ↑ | Recall ↑ |
|---|---|---|---|---|---|---|---|
| | 0.00 | 1.00 | 9.86 | 143.99 | 15.18 | 0.7486 | 0.5317 |
| | 0.25 | 1.00 | 5.21 | 181.20 | 8.33 | 0.7752 | 0.5542 |
| | 0.50 | 1.00 | 4.42 | 190.10 | **7.17** | 0.7872 | 0.5447 |
| | 0.75 | 1.00 | 4.09 | 201.98 | 8.06 | 0.7898 | 0.5448 |
| Later tokens | 1.00 | 2.50 | 4.70 | 185.14 | 7.74 | 0.7814 | 0.5543 |
| | 1.00 | 2.25 | 4.65 | 188.55 | 7.63 | 0.7857 | 0.5505 |
| | 1.00 | 2.00 | 4.67 | 188.11 | 7.74 | 0.7831 | 0.5514 |
| | 1.00 | 1.50 | 4.24 | 196.04 | 7.68 | 0.7901 | 0.5547 |
| All tokens | 1.00 | 1.00 | 3.91 | 207.92 | 8.12 | 0.7945 | 0.5547 |
| | 1.00 | 0.75 | 3.77 | 213.29 | 8.24 | 0.7958 | 0.5456 |
| | 1.00 | 0.50 | 3.53 | 220.55 | 8.28 | 0.8053 | **0.5488** |
| | 1.00 | 0.25 | 3.35 | 233.78 | 8.17 | 0.8099 | 0.5382 |
| Early tokens | 1.00 | 0.00 | **3.27** | **244.23** | 7.96 | **0.8234** | 0.5216 |
| | 1.25 | 0.00 | **3.27** | 240.35 | 7.94 | 0.8175 | 0.5226 |
| | 1.50 | 0.00 | 3.33 | 240.59 | 7.82 | 0.8188 | 0.5173 |

The results are summarized in Table 1. We observe the following key trends:

**(1) Strong late weighting** ($\alpha = 0, \beta = 1$) performs poorly across all metrics. Over-emphasizing late tokens encourages the model to defer most of the work to the end of the sequence, amplifying error accumulation—a well-known failure mode in AR generation (Arora et al., 2022; Ren et al., 2025b).

**(2) Mild late weighting** ($\alpha = 0.5, \beta = 1.0$) boosts sFID but degrades FID. Interestingly, this setting attains the best sFID (**7.17**), suggesting that lightly favoring later tokens can enhance local spatial fidelity even if overall perceptual quality declines.

**(3) Flat weights** ($\alpha = \beta = 1$) provide a strong and stable baseline, yielding balanced performance across metrics and particularly solid recall. This indicates that uniform weighting remains a robust default when no positional preference is imposed.

**(4) Strong early weighting** ($\alpha = 1.0, \beta = 0$) delivers further gains in FID and IS, with $(1.0, 0.0)$ achieving the best FID (**3.27**) and IS (**242.67**). Prioritizing early tokens appears to improve global structure and semantic consistency, leading to higher overall image quality.

**Takeaway:** These results reveal tokens at different ordinal positions also serve distinct roles during the early training stage. When $\beta > \alpha$, the model favors sharpness and realism (low FID, high IS) but may lose structural consistency (higher sFID). In contrast, $\alpha > \beta$ improves sFID but slightly harms overall quality. Mild early emphasis ($\alpha = 1.0, \ \beta = 0.0$) achieves the best FID and IS, while $(0.5, \ 1.0)$ yields the best sFID, making it ideal for tasks needing fine-grained alignment.

## 5 ORDINAL-BIASED RANDOM TRAINING: A CURRICULUM VIEW.

The ordinal asymmetry identified in Sec. 4 can be naturally framed within the perspective of curriculum learning, where models typically begin with easier examples before progressing to harder ones. Under randomized paths, earlier and later tokens have distinct functionalities, with earlier tokens focus on global structure, while later tokens refines local details. Therefore, it is intuitive to decouple the learning of earlier and later tokens into separate stages, following a curriculum learning strategy.

Building on this intuition, we propose **Ordinal-biased Random Training** (ORT), which instantiates the curriculum strategy via the *Ordinal Focal Loss* (Sec. 4.2). Specifically, we leverage two parameters, $(\alpha, \beta)$ to control the relative emphasis on early vs. late positions. Unless otherwise stated, we use $(\alpha, \beta) = (0.5, 1.0)$ for ablation following the varying of Table 1, which biases training toward later tokens while retaining coverage of earlier ones. This design ensures that ORT introduces no architectural changes and remains fully compatible with existing AR frameworks.

---

**Algorithm 1:** Ordinal-biased Random Training (ORT)

---

**Input:** Dataset $\mathcal{D}$, model $\theta$, epochs $E_1$ (random), $E_2$ (mixed), $E_3$ (raster)

**for** $epoch = 1$ **to** $E_1 + E_2 + E_3$ **do**
    **if** $epoch \leq E_1$ **then**
        $(\alpha, \beta) \leftarrow (0.75, 1.25)$ or $(1, 0)$, $\pi \leftarrow$ RANDOMORDER() ;          `// ordinal-biased`
        Apply $\omega(t; \alpha, \beta)$ (Eq. 5); update $\theta$ with $\sum_t \omega(t)\ell_t$ ;
    **else if** $epoch \leq E_1 + E_2$ **then**
        $k \leftarrow epoch - E_1$;   $u \leftarrow k/E_2$;   $p_{\text{rand}} \leftarrow 1 - u$ ;        `// linear decay 1→0`
        Sample $b \sim$ Bernoulli($p_{\text{rand}}$) ;          `// b=1: random, b=0: raster`
        **if** $b = 1$ **then**
            $(\alpha, \beta) \leftarrow (0.75, 1.25)$ or $(1, 0)$;   $\pi \leftarrow$ RANDOMORDER() ;     `// ordinal-biased`
            Compute $\ell_t$; apply $\omega(t; \alpha, \beta)$; update $\theta$ ;
        **else**
            $(\alpha, \beta) \leftarrow (1.0, 1.0)$;   $\pi \leftarrow$ RASTERORDER() ;         `// uniform`
            Train with uniform weighting (Eq. 4); update $\theta$ ;
    **else**
        $(\alpha, \beta) \leftarrow (1.0, 1.0)$, $\pi \leftarrow$ RASTERORDER() ;         `// uniform`
        Train with uniform weighting (Eq. 4); update $\theta$ ;

**Output:** Trained autoregressive model $\theta$

---

ORT can be seamlessly integrated into randomized-order AR training schedules. We simply replace the uniform weighting in the randomized phase with the Ordinal Focal Loss (Eq. 5) and treat $(\alpha, \beta)$ as a hyperparameter controlling the degree of late bias. The overall procedure is identical to RAR (Yu et al., 2024b), except that the randomized phase requires only half the epochs to reach comparable convergence. Importantly, inference remains unchanged since ORT is purely a training strategy, making it a drop-in replacement for the random phase in existing pipelines. Algorithm 1 summarizes the procedure. In ORT, we divide the whole training process into three phases, *i.e*, random, mixed, and raster, which follows the core idea from easy to hard in curriculum learning.

## 6 EXPERIMENTS

### 6.1 IMPLEMENTATIONS AND SETUPS

**Model Settings:** We adopt RAR (Yu et al., 2024b) as our baseline, as it achieves state-of-the-art performance among pure autoregressive (AR) models on ImageNet-1K-256 (see Table 2. RAR also natively support randomized generation paths, making it well-suited for investigating the effects of generation order and token-wise loss reweighting. We use the pretrained tokenizer (Chang et al., 2022) following RAR. In addition, we also evaluate on Alitok (Wu et al., 2025), another SOTA AR image generator, as a second baseline to test the generalizability of our proposed ORT.

**Dataset and Evaluation:** Experiments are conducted on ImageNet-1K (Deng et al., 2009) at 256×256 resolution. We adopt both FID and sFID as the main metrics to evaluate global image quality and local detail following (Dhariwal & Nichol, 2021), as well as Inception Score (IS).

**Training Details:** All models are trained using AdamW (Loshchilov et al., 2017) ($\beta_1 = 0.9$, $\beta_2 = 0.96$), with a learning rate of $4e-4$ weight and decay of $0.03$. The path annealing starts at $125,000$ iterations and ends at $187,500$ iterations. The batch size of ablations (Table 1) is 256 on 4 A100 GPUs. The batch size of main results (Sec. 6.2) is 2048 on 32 A800 GPUs.

### 6.2 MAIN RESULTS

Table 2 compares different generative paradigms on ImageNet-1K $256 \times 256$. Our RAR-XL baseline reaches FID 1.50 after 400 epochs. We use a slightly stronger setting $(\alpha, \beta) = (0.75, 1.25)$ for $\text{ORT}_l$ and $(\alpha, \beta) = (1, 0)$ for $\text{ORT}_e$, which consistently yields stable convergence and competitive FID. With the proposed ORT, we achieve a comparable FID of $1.53$ but with only 300 epochs of training, reducing the randomized training stage by half. This demonstrates that ORT accelerates

Table 2: **Comparison on ImageNet-1K 256×256.** We report FID↓, IS↑, sFID↓, Precision (Prec.↑), and Recall (Rec.↑) using the ADM evaluation protocol. Diffusion, Masked Diffusion, Variational Autoregressive (VAR), Multimodal AR (MAR), FlowAR, and Autoregressive (AR) methods are grouped. Our RAR baselines establish strong AR performance, while integrating ORT further reduces training epoch and keeps the advances to diffusion-based models. * denotes our implementation.

| Type | Generator | Venue | #Params | #Epoch | FID↓ | IS↑ | sFID↓ | Pre.↑ | Rec.↑ |
|---|---|---|---|---|---|---|---|---|---|
| Diff. | LDM-8 (Rombach et al., 2022) | CVPR 22 | 258M | - | 7.76 | 209.5 | - | 0.84 | 0.35 |
| Diff. | LDM-4 (Rombach et al., 2022) | CVPR 22 | 400M | - | 3.60 | 247.7 | - | 0.87 | 0.48 |
| Diff. | UViT-L/2 (Bao et al., 2023) | CVPR 23 | 287M | 240 | 3.40 | 219.9 | - | 0.83 | 0.52 |
| Diff. | UViT-H/2 (Bao et al., 2023) | CVPR 23 | 501M | 400 | 2.29 | 263.9 | - | 0.82 | 0.57 |
| Diff. | DiT-XL/2 (Peebles & Xie, 2023) | ICCV 23 | 675M | 1400 | 2.27 | 278.2 | 4.60 | 0.83 | 0.57 |
| Diff. | SiT-XL/2 (Ma et al., 2024) | ECCV 24 | 675M | 1400 | 2.06 | 270.3 | 4.50 | 0.82 | 0.59 |
| Diff. | DiMR-XL/2R (Liu et al., 2024) | NeurIPS 24 | 505M | 800 | 1.70 | 289.0 | - | 0.79 | 0.63 |
| Diff. | MDTV2-XL/2 (Gao et al., 2023) | ICCV 23 | 676M | 1000 | 1.58 | 314.7 | 4.52 | 0.79 | 0.65 |
| Diff. | REPA (Yu et al., 2025) | ICLR 25 | 675M | 800 | 1.42 | 305.7 | 4.70 | 0.80 | 0.65 |
| Mask. | MaskGIT (Chang et al., 2022) | CVPR 22 | 177M | 300 | 6.18 | 182.1 | - | 0.80 | 0.51 |
| Mask. | TiTok-S-128 (Yu et al., 2024c) | NeurIPS 24 | 287M | 300 | 1.97 | 281.8 | - | - | - |
| Mask. | MaskBit (Weber et al., 2024) | Arxiv 24 | 305M | 1080 | 1.52 | 328.6 | - | - | - |
| VAR | VAR-d30 (Tian et al., 2024) | NeurIPS 24 | 2.0B | 350 | 1.92 | 323.1 | - | 0.82 | 0.59 |
| VAR | VAR-d30-re (Tian et al., 2024) | NeurIPS 24 | 2.0B | 350 | 1.73 | 325.0 | - | 0.82 | 0.60 |
| MAR | MAR-B (Li et al., 2024) | NeurIPS 24 | 208M | 800 | 2.31 | 281.7 | - | 0.82 | 0.57 |
| MAR | MAR-L (Li et al., 2024) | NeurIPS 24 | 479M | 800 | 1.78 | 296.0 | - | 0.81 | 0.61 |
| MAR | MAR-H (Li et al., 2024) | NeurIPS 24 | 943M | 800 | 1.55 | 303.7 | - | 0.81 | 0.62 |
| FlowAR | FlowAR-S (Ren et al., 2025a) | ICML 25 | 170M | 400 | 3.61 | 234.1 | - | 0.83 | 0.50 |
| FlowAR | FlowAR-H (Ren et al., 2025a) | ICML 25 | 1.9B | 400 | 1.65 | 296.5 | - | 0.83 | 0.60 |
| AR | GPT2 (Esser et al., 2021a) | CVPR 21 | 1.4B | - | 15.78 | 74.3 | - | - | - |
| AR | GPT2-re (Esser et al., 2021a) | CVPR 21 | 1.4B | - | 5.20 | 280.3 | - | - | - |
| AR | VIM-L (Yu et al., 2022) | ICLR 22 | 1.7B | - | 4.17 | 175.1 | - | - | - |
| AR | VIM-L-re (Yu et al., 2022) | ICLR 22 | 1.7B | - | 3.04 | 227.4 | - | - | - |
| AR | Open-MAGVIT-B (Luo et al., 2024) | Arxiv 24 | 343M | - | 3.08 | 258.3 | - | 0.85 | 0.51 |
| AR | Open-MAGVIT2-L (Luo et al., 2024) | Arxiv 24 | 804M | - | 2.51 | 271.7 | - | 0.84 | 0.54 |
| AR | Open-MAGVIT2-XL (Luo et al., 2024) | Arxiv 24 | 1.5B | 350 | 2.03 | 286.0 | - | 0.84 | 0.54 |
| AR | LlamaGen-L (Sun et al., 2024) | Arxiv 24 | 343M | 300 | 3.07 | 256.1 | - | 0.83 | 0.52 |
| AR | LlamaGen-3B (Sun et al., 2024) | Arxiv 24 | 3.1B | 300 | 2.18 | 263.3 | - | 0.81 | 0.58 |
| AR | RandAR-L (Pang et al., 2025) | CVPR 25 | 343M | 300 | 2.55 | 288.8 | - | 0.81 | 0.58 |
| AR | RandAR-XXL (Pang et al., 2025) | CVPR 25 | 1.4B | 300 | 2.15 | 322.0 | - | 0.79 | 0.62 |
| AR | RAR-B | ICCV 25 | 261M | 400 | 1.95 | 290.5 | - | 0.82 | 0.58 |
| AR | RAR-L | ICCV 25 | 461M | 400 | 1.70 | 299.5 | - | 0.81 | 0.60 |
| AR | RAR-XL | ICCV 25 | 955M | 400 | 1.50 | 306.9 | - | 0.80 | 0.62 |
| AR | Alitok-XL* (Wu et al., 2025) | Arxiv25 | 662M | 400 | 1.37 | 321.4 | 7.29 | 0.79 | 0.64 |
| AR | RAR-B+ORT$_l$ (ours) | - | 261M | 300 | 1.97 | 275.7 | 6.05 | 0.81 | 0.58 |
| AR | RAR-L+ORT$_l$ (ours) | - | 461M | 300 | 1.71 | 293.6 | 5.63 | 0.81 | 0.60 |
| AR | RAR-XL+ORT$_l$ (ours) | - | 955M | 300 | 1.53 | 299.5 | 5.27 | 0.81 | 0.60 |
| AR | Alitok-XL*+ORT$_l$ (ours) | - | 662M | 300 | 1.34 | 309.1 | 7.15 | 0.79 | 0.65 |
| AR | Alitok-XL*+ORT$_e$ (ours) | - | 662M | 300 | **1.26** | 297.9 | 7.56 | 0.78 | 0.66 |

convergence significantly without sacrificing final generation quality, highlighting the efficiency benefits of order-aware training.

Besides RAR, we also conduct experiments on Alitok. We observe ORT can lead to better FID score despite using 100 fewer epochs with Alitok, for both ORT$_e$ and ORT$_l$. For instance, ORT$_e$ achieves SOTA FID score of 1.26 on Alitok-XL with only 300 epochs, while the baseline Alitok-XL scores 1.37 FID at 400 epochs. This demonstrates that ORT is generalizable across different base models.

## 6.3 TIME COMPLEXITY ANALYSIS

ORT introduces almost no extra training cost, since the only modification relative to RAR is computing position-dependent weights $\omega(t; \alpha, \beta)$ in the ordinal focal loss (Eq. 5), which is a simple linear interpolation negligible compared to forward/backward passes. No additional parameters, memory, or sampling steps are required, and inference remains identical to the baseline. Empirically, we measure wall-clock time per iteration (batchsize is 2048) on ImageNet-256 with RAR-L and find the difference within measurement noise ( 0.03%), confirming that ORT 's efficiency gains stem entirely from faster convergence rather than reduced per-step cost.

Table 3: Per-batch training time (ms) on ImageNet-256 with RAR-L.

| Method | Time / batch | Overhead |
|---|---|---|
| RAR (baseline) | 0.660 s | – |
| ORT (ours) | 0.662 s | 0.3% |

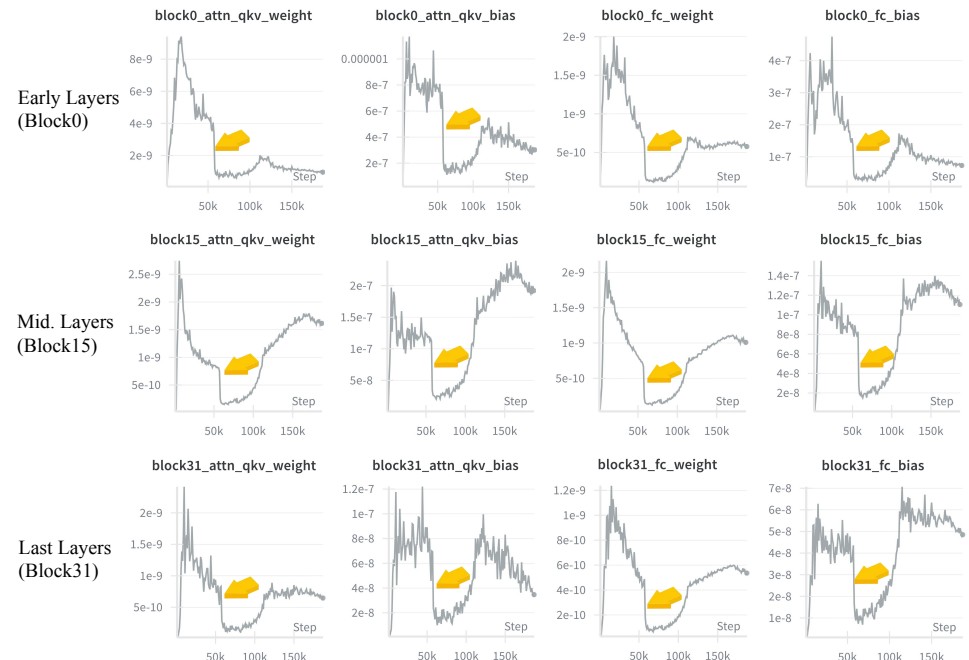

Figure 4: Gradient dynamics of RAR-XL with ORT ($\alpha = 0.5, \beta = 1$). Across layers of different depth (Block0, Block15, Block31) and parameter types (attention QKV weights/bias and feedforward weights/bias), the gradient norms remain large at the beginning but collapse sharply around the midpoint of training. Yellow arrows mark this *sudden convergence* phenomenon, consistently observed across modules. After convergence, the gradual increase in gradient norms corresponds to the annealing phase from random to raster training (see Alg. 1).

Table 4: Random Path with different $(\alpha, \beta)$ at $75k$ training steps.

| Path | $\alpha$ | $\beta$ | **FID** | **sFID** |
|------|------|------|------|------|
| Random | 1 | 1 | 9.72 | 11.46 |
| Random | 0.5 | 1 | 3.89 | 7.54 |
| Random | 0.75 | 1.25 | **3.84** | **7.22** |

Table 5: Raster Path with different $(\alpha, \beta)$ at $50k$ & $100k$ steps.

| Path | $\alpha$ | $\beta$ | **FID** | |
|------|------|------|------|------|
| | | | $50k$ | $100k$ |
| Raster | 1 | 1 | **3.75** | **2.83** |
| Raster | 0.5 | 1 | 3.76 | 2.94 |

Table 6: Comparison of RAR and ORT under same training epoch.

| **Strategy** | **#Epoch** | **FID** | **IS** | **sFID** |
|------|------|------|------|------|
| RAR | 300 | 2.04 | **284.5** | 6.10 |
| ORT$_l$ | 300 | **1.97** | 275.7 | **6.05** |
| RAR | 400 | **1.95** | 290.5 | - |
| ORT$_l$ | 400 | 1.96 | 286.3 | 6.03 |

## 6.4 ABLATIONS

**Ordinal Bias in Random Phase:** We first analyze the effect of different $(\alpha, \beta)$ settings during the randomized training phase (RAR-B in Table 4). At $75k$ steps—well before the full 200-epoch randomized schedule—we observe that ORT$_l$ with $(\alpha = 0.75, \beta = 1.25)$ significantly accelerates convergence, achieving FID 3.89, while the baseline $(\alpha = 1, \beta = 1)$ scores merely 9.72 FID. This confirms that emphasizing later tokens indeed facilitates faster convergence in the random phase.

**Ordinal Bias in Raster Phase:** Next, we evaluate the effect of ordinal-biased weighting under raster paths (Table 5). Unlike in the randomized setting, applying non-uniform weights here leads to degraded performance (FID 3.75 vs. 3.76), indicating that raster training requires uniform supervision. We provide a more detailed discussion of this phenomenon in Appendix A.

**Faster Convergence with ORT:** Finally, we report the end-to-end results of full ORT training (Table 6) with $(\alpha = 0.75, \beta = 1.25)$. With our three-stage schedule, ORT reaches an FID of 1.97 and sFID of 6.05 at 300 epochs, outperforming the baseline (2.04 / 6.10) while reducing the randomized phase by half, thereby validating its efficiency advantage in large-scale training. In addition, to ablate the effect of overall training epochs (e.g. learning rate scheduling), in Table 6 we also compare ORT and baseline RAR trained end-to-end with 300 and 400 epochs. We can see despite baseline and ORT$_l$ have comparable performance under 400 epochs, ORT$_l$ performs baseline notably at 300 epochs.

Table 7: FID of RAR-B baseline with ORT ($\alpha = 0.5, \beta = 1$). ORT shows a sudden convergence at around $60k$ steps.

| Iter | Grad Norm Trend | FID |
|------|-----------------|-----|
| $50k$ | Large, unstable | 17.042 |
| $75k$ | Sudden Converged | 3.89 |

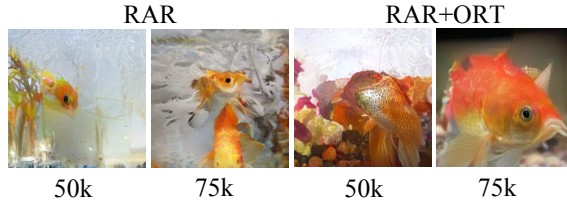

RAR       RAR+ORT

50k    75k    50k    75k

Figure 5: Qualitative samples at 50k and 75k iterations of RAR-B with ORT ($\alpha = 0.5, \beta = 1$).

Table 8: Training and tuning cost for ORT vs. the 400-epoch RAR-XL baseline on ImageNet-256. Each ORT run saves about 445 H800 GPU-hours compared to the baseline. The one-time tuning of $(\alpha, \beta)$ is performed on the smaller RAR-B model using 5 short 50-epoch runs on 4×A100 (600 GPU-hours total), and is reused for all main experiments. At a cloud price of \$1.5 per H800-hour, 445 H800 GPU-hours $\approx$ \$670 saved per full RAR-XL run, so the tuning cost is quickly amortized after a few runs.

| Stage | Setup | Wall-Time | GPU-Hours | Notes |
|-------|-------|-----------|-----------|-------|
| RAR-XL baseline training (400 ep) | 32×H800 | 55.5 h | 1776 | reference |
| ORT training (300 ep, per run) | 32×H800 | 41.6 h | 1331 | saves 445 H800-h ($\approx$25%) |
| ORT hyperparam tuning (5×50 ep on RAR-B) | 4×A100 | 5×30 h = 150 h | 600 | one-time cost |

**Sudden Convergence of ORT:** A distinctive phenomenon of ORT is a *sudden convergence* during the randomized phase. As shown in Fig. 4, gradient norms stay large early on but collapse sharply around 100 epochs (62.5k steps), a transition absent in the uniform RAR baseline. This indicates that late-biased weighting exploits easier late tokens to accelerate early optimization, allowing the model to stabilize much faster. Table 7 and Fig. 5 further show that this collapse coincides with a sharp FID drop under ORT, confirming that the effect reflects faster random-phase convergence rather than altered final performance.

**Training Efficiency and Cost Analysis:** To quantify the practical benefits of ORT, Table 8 compares its compute footprint with the original 400-epoch RAR-XL schedule on ImageNet-256. Training RAR-XL for 300 epochs with ORT requires 41.6 hours on 32×H800 GPUs, whereas the 400-epoch baseline requires 55.5 hours, corresponding to an additional 445 GPU-hours. At a realistic cloud rate of 1.5 per H800-hour, this translates to approximately 670 saved per full training run, while yielding essentially identical final performance (gFID 1.50 vs. 1.53).

Moreover, ORT introduces only minimal tuning overhead. We determine the curriculum parameters $(\alpha, \beta)$ once on the smaller RAR-B model using a coarse sweep over just five recommended candidates—$(1, 0)$, $(0.25, 1)$, $(0.5, 1)$, $(0.75, 1)$, and $(0.75, 1.25)$. Each 50-epoch RAR-B run takes about 30 hours on 4×A100 ($\approx$ 120 GPU-hours), so the entire sweep costs only 600 A100 GPU-hours and is incurred a single time. The resulting $(\alpha, \beta)$ setting transfers directly to RAR-XL and all main experiments without any further tuning. Since ORT saves 445 H800 GPU-hours per large-scale run, this one-time sweep is quickly amortized.

Overall, ORT provides a practically significant efficiency gain: it reduces training cost, avoids repeated hyperparameter searches, and maintains final generation quality, making it attractive for both academic labs and large-scale deployments.

## 6.5 VISUALIZATION

To qualitatively evaluate the impact of our training strategy, we visualize samples generated by RAR-B trained with $\mathrm{ORT}_l$ on ImageNet-256. As shown in Fig. 6, RAR-B trained with $\mathrm{ORT}_l$ generates high-quality samples with coherent global structures and realistic local details, illustrating the effectiveness of our training strategy.

## 7 LIMITATION AND FUTURE WORK

While ORT is simple and effective, it still has limitations. First, its performance can depend on the choice of weighting parameters $(\alpha, \beta)$: as shown in Table 1, the relative emphasis on early vs. late

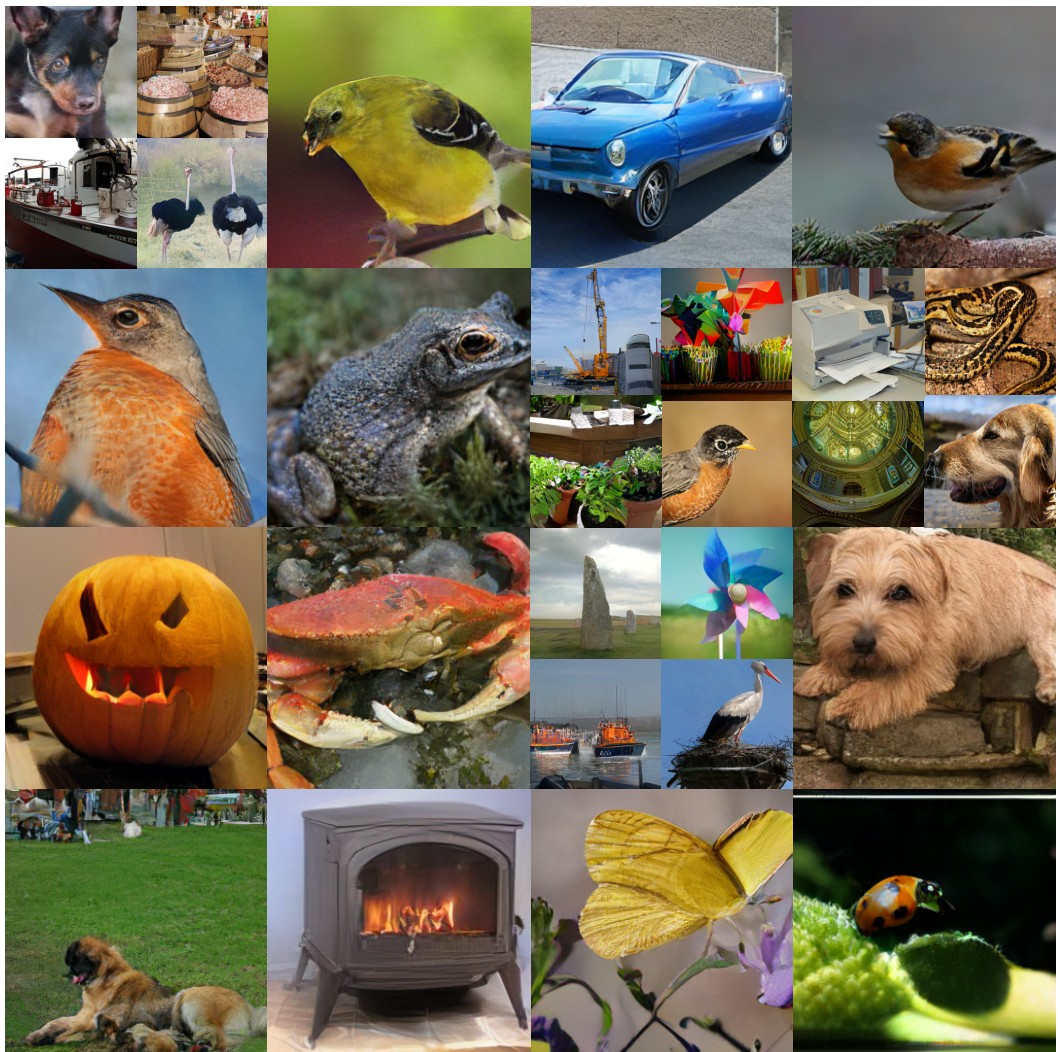

Figure 6: Qualitative results on ImageNet-256 using RAR-B with $\text{ORT}_l$.

tokens influences convergence in the early phase, and different AR architectures may require different settings. Second, our current ordinal focal loss adopts a fixed linear interpolation, without exploring alternative schedules (*e.g.*, cosine) or adapting to token-specific difficulty or confidence. Future work could design adaptive weighting strategies that automatically adjust across token positions, potentially improving both efficiency and robustness.

## 8  CONCLUSION

In this work, we revisited randomized-path training for visual autoregressive models and revealed a consistent *ordinal asymmetry*: early tokens are harder to optimize due to minimal context, while later tokens converge more easily with richer context. Building on this observation, we introduced **Ordinal-biased Random Training** (ORT), a simple curriculum strategy that emphasizes one subproblem in the early phase: either early tokens or later tokens. ORT shortens the randomized training stage by half, induces a distinctive sudden convergence phenomenon in gradient dynamics, and achieves comparable generation quality to the full baseline without altering inference. These findings highlight that ordinal positions play distinct functional roles during autoregressive training and that embracing rather than suppressing this asymmetry can yield substantial efficiency gains.

ETHICS STATEMENT

This work uses only publicly available datasets and does not involve human subjects or sensitive data. While autoregressive image generation models may be misused for harmful content, our contributions do not increase these risks beyond existing methods. We encourage responsible use and adherence to dataset and model release guidelines.

REPRODUCIBILITY STATEMENT

We describe all model settings, training schedules, and evaluation protocols in detail in Sec. 6.2. Hyperparameters and ablation settings are reported in the main text and appendix. We commit to releasing core code and scripts for ORT after the review process to facilitate reproducibility.

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

## A APPENDIX 1: WHY LATE-TOKEN CONTRIBUTIONS EMERGE ONLY UNDER RANDOMIZED PATHS

**Definition 1 (Ordinal Information Potential).** We define the *Ordinal Information Potential (OIP)* of token $x_{\pi_t}$ along a path $\pi$ as the expected KL divergence between the model's conditional prediction and its marginal distribution:

$$\Phi_t^\pi = \mathbb{E}_{x_{\pi(<t)}} \left[ \mathrm{KL} \left( p(x_{\pi_t} \mid x_{\pi_{<t}}) \, \| \, p(x_{\pi_t}) \right) \right]. \tag{7}$$

Intuitively, a higher OIP indicates that the model needs to introduce more information at this step to reduce uncertainty, typically associated with generating global structure. In contrast, a lower OIP implies that the token prediction is already well-guided by the context, often focusing on local refinement.

**Proposition 1** *OIP Monotonicity under Random Paths. Let $\pi$ be a uniformly random generation order. If (i) the model effectively exploits context, and (ii) the context tokens provide non-redundant information, then the Ordinal Information Potential $\Phi_t^\pi$ decreases with time step $t$:*

$$t_1 < t_2 \quad \Rightarrow \quad \Phi_{t_1}^\pi \geq \Phi_{t_2}^\pi$$

**Proof.** Under a random generation order, the expected context size $|\pi(<t)|$ increases with $t$. If the model effectively utilizes this context to make more confident predictions, then the conditional distribution $p(x_{\pi_t} \mid x_{\pi(<t)})$ becomes increasingly peaked (*i.e.*, low entropy), concentrating mass around the correct token. Since the marginal distribution $p(x_{\pi_t})$ is fixed, the KL divergence from this peaked conditional to the marginal decreases, reflecting reduced uncertainty and higher predictability. Therefore, under the assumption of effective modeling and non-redundant context, $\Phi_t^\pi$ tends to decrease with $t$.

**Remark 2** *Random Paths Focus More on Details. For two paths $\pi_{raster}$ and random path $\pi$ of the same length, if the context $x_{\pi_{<t}}$ of $\pi$ covers a more isotropic spatial neighborhood around $x_{\pi_t}$, we have:*

$$\Phi_t^{\pi_{random}} < \Phi_t^{\pi_{raster}}.$$

Under the assumption, the model effectively utilizes a spatially diverse context.

**Justification.** In raster paths, context tokens before step $t$ are spatially biased—typically aligned along top-left to bottom-right—providing uni-directional and incomplete spatial information. This directional limitation constrains the mutual information $I(x_{\pi_t}; x_{\pi_{<t}})$, as many local cues (e.g., symmetric patterns, object closure, textures) are missing. In contrast, random paths tend to provide context from all directions around $x_{\pi_t}$, allowing the model to make more confident and spatially consistent predictions. This leads to faster entropy reduction and smaller OIP, implying more deterministic generation suitable for local refinement. Therefore, even if both paths have the same length, random paths exhibit stronger refinement capacity in later steps due to their isotropic contextual suppORT.

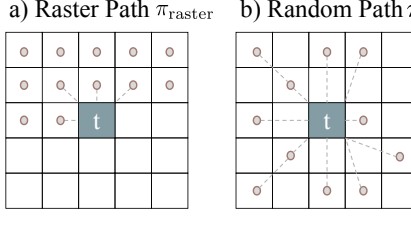

a) Raster Path $\pi_{\mathrm{raster}}$    b) Random Path $\pi$

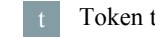

○ Visual Information    [t] Token t

Figure 7: Contextual Differences Between Generation Paths. Random path (b) offers more global visual information than raster path (a), facilitating details predictions.

## B APPENDIX 2: LLM USAGE

In preparing this manuscript, we made limited use of large language models (LLMs) such as ChatGPT. The models were employed solely for linguistic assistance, including polishing phrasing, improving readability, and detecting grammar or spelling errors. No experimental design, implementation, analysis, or result generation relied on LLMs. All technical content, algorithms, and experimental findings were conceived and validated by the authors.

