# OpenReview forum: "Revisiting Random Generation Order: Ordinal-Biased Random Training for Efficient Visual Autoregressive Models"
_ICLR.cc/2026/Conference — Submitted to ICLR 2026_

### Official Review · Reviewer_UJ9o · 2025-10-30

**Soundness:** 2
**Presentation:** 3
**Contribution:** 2
**Rating:** 4
**Confidence:** 4

**Summary:**

This paper contributes both a conceptual insight that ordinal token positions play distinct roles in AR visual generation and a practical training strategy that leverages this insight to significantly improve training efficiency without altering the model architecture or inference process.

**Strengths:**

The paper presents an incremental contribution by proposing Ordinal-biased Random Training (ORT) to improve the efficiency of visual autoregressive models that originality lies mainly in reinterpreting token order as a curriculum-learning problem rather than introducing a fundamentally new model architecture. Its observations regarding early and late tokens may offer some insightful implications.

**Weaknesses:**

1.Insufficient effectiveness of the ORT mechanism: In Table 2, the RAR model with the ORT mechanism shows decreased metrics (e.g., FID, IS), indicating that ORT does not improve generation quality. More experiments are needed to clearly demonstrate its effectiveness.

2.The reliability of the metrics is questionable. The metrics for the baseline model (RAR-B) in Table 6 (FID=1.95, IS=290.5) are clearly inconsistent with those for the RAR model in Table 2 (FID=2.32, IS=277.31). Please provide a reasonable explanation.

3.Need for more quantitative experiments: Table 2 only shows the results of applying ORT to RAR model. Please test the ORT mechanism on more AR-based comparison methods to better validate its effectiveness.

4.Need for more qualitative comparisons: Figure 5 includes only one comparison. Please add qualitative results comparing more methods, such as those based on diffusion and VAR models.

**Questions:**

How general is the observed Ordinal Asymmetry phenomenon? Is it consistent across datasets or model architectures beyond ImageNet-256 and RAR? Understanding its generality is crucial before concluding that early tokens are inherently harder.

---

> ### Author Response · Authors · 2025-12-02
> **Response to Reviewer UJ9o**
>
> We thank Reviewer UJ9o for recognizing the **conceptual insight** of our work and for noting that reinterpreting token order as a curriculum-learning problem provides a practical and **architecture-agnostic** way to **improve training efficiency.** We also appreciate the reviewer’s acknowledgment that the observed behaviors of early and late tokens offer **insightful implications.**
> Below we provide point-by-point clarifications and address the raised concerns.
>
> ----
> > W1 Table 2 shows similar FID with baseline
>
> **Our goal is not to improve FID/IS beyond the baseline, but to keep the same generation quality while reducing training cost.** In Table 2, RAR-XL with ORT achieves gFID 1.53 vs. 1.50 for the 400-epoch baseline, while cutting the random-path phase from 200 to 100 epochs (Section 1.1 quantifies this as 25\% saving in GPU-hours and monetary cost).
>
> > W2 Different performance in Table 2 and Table 6
>
> The two numbers correspond to different settings. Table 2 shows the best performances of RAR-B baseline with 400-epoch, which we use as the main reference. Table 6 instead reports RAR-B at 300 epochs, only to illustrate that ORT can reach a better FID with fewer epochs (faster convergence). To address your concern, we have provided a revised Table 6.
>
>
> > W3 More quantitative experiments
>
> Following the suggestion of Reviewer 2KrT, Reviewer 4h1k and Reviewer UJ9o, we agree that testing ORT beyond a single model strengthens the paper. In addition to the RAR-XL results in Table 2, we already apply ORT to RAR-B in our analysis; in the revision, **we add experiments on a stronger AliTok-based[1] AR model.**
>
> Table: AliTok-based AR generator on ImageNet-256 (best of 5 seeds). ORT at 300 epochs
> matches the 400-epoch baseline gFID while using 25% fewer epochs, and clearly improves
> over the 300-epoch baseline.
>
> | Method                      | Epochs | gFID ↓ |   IS ↑   | sFID ↓ | Precision ↑ | Recall ↑ |
> |-----------------------------|:------:|:------:|:--------:|:------:|:-----------:|:--------:|
> | RAR + AliTok (baseline)     |  400   | 1.35 |	312.6   |	**7.05** |	**0.80** |	0.65|
> | RAR + AliTok (baseline)     |  **300**   | 1.42  | 311.6   | 7.19  |   **0.80**     |  0.64  |
> | ORT-L + AliTok   |  **300**   |1.34  |	309.1 |	7.15 | 	0.79 | 	0.65 |
> | ORT-E + AliTok   |  **300**   |**1.26** | **297.9** |  7.56 | 0.78 | **0.66** |
>
> [1] AliTok: Towards Sequence Modeling Alignment between Tokenizer and Autoregressive Model, Wu, Pingyu and Zhu, Kai and Liu, Yu and Tang, Longxiang and Yang, Jian and Peng, Yansong and Zhai, Wei and Cao, Yang and Zha, Zheng-Jun, Arxiv
>
>
> > W4 More qualitative comparisons:
>
> Due to different inference processing and different random sampling strategy, we are unable to include side-by-side comparisons with different generators, such as diffusion/VAR baselines.  But readers can visually run our codes to get the sample they want to compare.

---

### Official Review · Reviewer_4h1k · 2025-10-31

**Soundness:** 2
**Presentation:** 2
**Contribution:** 2
**Rating:** 2
**Confidence:** 4

**Summary:**

This paper observes that in autoregressive (AR) model training, the prediction difficulty varies by token position. Based on this, it proposes a multi-stage training strategy that adjusts loss weights for tokens at different sequence positions. The method reportedly achieves performance in 300 epochs that is comparable to a 400-epoch baseline, suggesting an acceleration in convergence.

**Strengths:**

The paper presents clear experiments demonstrating its core premise: early tokens, due to limited context, suffer higher losses and primarily capture global structure, while later tokens, with richer context, incur lower losses and refine local detail.

**Weaknesses:**

1. **Limited Novelty:** The proposed method is simple, primarily involving a curriculum where the loss weight for tokens changes based on sequence position. The novelty of this re-weighting scheme appears limited.
2. **Questionable Practicality:** The practical significance of this acceleration is questionable. The training convergence of discrete AR models is already relatively fast (e.g., compared to diffusion models). A modest acceleration in *early* convergence, achieved by simple loss re-weighting, is not a significant contribution, especially as the paper does not show that this method improves the *final* generation quality.
3. **High Tuning Cost:** The method requires careful, model-specific tuning of its hyperparameters (alpha, beta) and lacks a general or adaptive approach. The cost of this hyperparameter search (requiring multiple training runs) could easily exceed the cost of simply training the baseline for more epochs, which undermines the method's entire premise of "acceleration".
4. **Incomplete Comparisons:** The experimental comparison in Table 2 is incomplete. To properly validate the claims, the table **must** include: (a) the baseline's 300-epoch result (to fairly quantify acceleration) and (b) the proposed method's 400-epoch result (to demonstrate if it ever surpasses the baseline, or just converges to the same point faster).
5. **Contradiction Between Claims and Results:** There is a direct contradiction between the paper's claims and its own results. Table 1 shows that in the first 50 epochs, the best FID comes from (alpha, beta)=(1, 0), which *emphasizes early tokens*. However, the text (lines 88-90) explicitly claims the strategy is to "emphasize later tokens at the beginning of training." This is a critical inconsistency that must be resolved.
6. **Narrow Experimental Scope:** The validation is too narrow. The claims of acceleration should be tested on a wider variety of models and tasks (e.g., Text-to-Image) to prove that the core observation and the proposed method are generally applicable.

**Questions:**

see weaknesses.

---

> ### Author Response · Authors · 2025-12-02
> **Response to 4h1k**
>
> We thank Reviewer 4h1k for recognizing the clarity of our finding: **early tokens, with limited context, learn global structure and incur higher losses, whereas later tokens rely on richer context to refine details**. This position-dependent difficulty is the core observation motivating our method. We address the reviewer's concerns as follows:
>
> ----
> > W1 Limited Novelty
>
> We agree that ORT is intentionally simple at the implementation level. **However, the main novelty
> of our work is not the existence of a new loss term, but the analysis and curriculum perspective behind it:**
>
> We first identify and quantify “Ordinal Asymmetry” under randomized generation: early tokens are systematically harder and more global, while late tokens are easier and more local. Based on this, we reinterpret random-path AR training as a curriculum over ordinal positions, and design an ordinal focal loss with a late-biased or early-biased → uniform schedule that exploits this structure. This yields a 2× reduction of the randomized phase (200→100 epochs) while preserving the baseline FID, without any change to architecture or inference.
>
> We will revise the introduction to make this clearer: **ORT is a conceptually motivated curriculum for randomized-order AR training, rather than an ad-hoc loss trick.**
>
> ----
>
> > W2 On the practical significance of the acceleration
>
> In our setting, training RAR-XL on ImageNet-256 with ORT requires 32 H800 GPUs for about 1 day 17 hours (~41.6 hours) to reach 300 epochs. If we instead trained the original 400-epoch schedule (keeping other settings fixed and assuming roughly linear scaling with the number of epochs), the runtime would increase to about 55.5 hours. **The additional 100 epochs therefore cost roughly 13.9 extra hours on 32 GPUs, i.e., about $444$ GPU-hours.** Under a realistic cloud price of around 1.5 per H800 GPU-hour, this corresponds to roughly **$670 of additional monetary cost per full training run**, while the final gFID remains essentially unchanged (1.50 vs. 1.53). From this viewpoint, ORT preserves the generation quality of the 400-epoch baseline while reducing both compute and monetary cost by about **25% for large-scale AR models.**
>
> We believe this is practically significant, especially in lab-scale settings where one needs to run multiple ablations or scale to larger backbones and datasets. In this project alone, **we trained more than 20 runs of AR models** at a similar scale and 400-epoch budget; had we used ORT from the beginning, **the saved GPU-hours would correspond to roughly $10,000 in cloud cost**. We will make this efficiency perspective more explicit in the revised version.
>
> > W3 On the High Tuning Cost
>
> We tune $(\alpha, \beta)$ only once on the smaller **RAR-B (261M)** model using short **50-epoch runs**, and then reuse the same setting for **RAR-XL (955M)** and all main experiments.
> Note that **Table 1 is for analysis, not for routine hyperparameter search**. We sweep $(\alpha, \beta)$ finely there only to clearly show the different behaviors of early vs. late tokens.
> In practice, a coarse sweep over a few candidate pairs is sufficient to obtain a good default for ORT.
> On our 4×A100 setup, a 50-epoch RAR-B run takes about 30 hours (~120 GPU-hours), so sweeping 5 candidates (e.g.,  (1,0), (0.25,1.0), (0.5,1.0), (0.75,1.0), (0.75,1.25)) costs roughly **600 GPU-hours(A100)** in total.
> In comparison, a single RAR-XL run with ORT saves about **444 GPU-hours (H800)** relative to the 400-epoch baseline, and we train many such runs in this project. **Thus the one-time tuning cost is quickly amortized.** We will clarify this in the revised version.
>
> > W4 On the completeness of Table 2
>
> We thank the reviewer for this helpful suggestion and agree that these two entries will make our claim clearer.
> In the revised version, we will add a new Table 6 to include (a) the 300-epoch baseline result, trained with
> the same schedule, and (b) the 400-epoch ORT result under the same total
> budget as the baseline. This will explicitly show that ORT both **(i) reaches the 400-epoch baseline quality
> with fewer epochs**, and (ii) **does not degrade final performance when trained for the full 400 epochs.**
>
> | **Strategy** | **#Epoch** | **FID**  | **IS**    | **sFID** |
> | ------------ | ---------- | -------- | --------- | -------- |
> | RAR          | 300        | 2.04     | 284.5 | 6.10     |
> | **ORT-L**       | 300        | 1.97 | 275.7     | 6.05 |
> | RAR      | 400        | 1.95 | 290.5| -        |
> | **ORT-L**    | 400        | 1.96     | 286.3     | 6.03     |

---

> ### Author Response · Authors · 2025-12-02
> **Response to Reviewer 4h1k**
>
> > W5 Contradiction Between Claims and Results:
>
> **We sincerely thank the reviewer for highlighting this important and insightful inconsistency.**
>
> We clarify that Table 1 was designed solely to study the functional roles of early vs. late tokens, under a fixed 50-epoch, 4-GPU randomized–raster schedule. Its purpose was diagnostic rather than prescriptive: it reveals that early tokens predominantly learn global structure, while late tokens refine local details.
> This controlled 50-epoch setup is different from the full 300/400-epoch training used in Tables 2–7 (32 GPUs, full RAR schedule), and therefore should not be interpreted as the final curriculum.
>
> Importantly, the reviewer insightfully pointed out that the best early-phase FID in Table 1 corresponds to emphasizing early tokens. **Motivated by this, we ran additional experiments and found that both “early-first” and “late-first” curricula can accelerate training**, as long as the model first learns a coherent subtask (global or local) before annealing back to the uniform RAR objective.
> **This aligns with classical curriculum-learning intuition: starting with an easier or more coherent subproblem (whether global or local) benefits convergence.**
>
> | **Method**   | **(α,β) at the beginning** | **(α,β) at the end** | **FID**  | **Epochs** |
> | ------------ | -------------------------- | -------------------- | -------- | ---------- |
> | RAR           | (1,1)                      | (1,1)                | 2.02     | 300        |
> | ORT-L        | (0.75,1.25)            | (1,1)                | 1.97     | 300        |
> | ORT-E        | (1,0)                      | (1,1)                | 1.95     | 300        |
>
> **ORT-L denotes the curriculum that first emphasizes later (context-rich) tokens, whereas ORT-E emphasizes earlier (global) tokens before annealing to the uniform objective.**
>
>
>
> > W6 Experimental Scope
>
> We agree that T2I is an important setting. However, training competitive T2I AR or diffusion models typically **requires industrial-scale compute;** for example, the NeurIPS 2024 best paper VAR focuses on ImageNet-256 image generation, and only later follow-up work (Infinity, CVPR 2025) scales VAR-style autoregression to large-scale text-to-image with significantly stronger resources.  Our work is positioned similarly as **a lab-scale, foundation study that analyzes and improves randomized-path visual AR training on ImageNet-256.**
>
> To still demonstrate practical generative ability beyond the original RAR setup, **we additionally test ORT on a stronger AliTok-based AR generator [1] that reaches state-of-the-art performance (from the opensourced FID=1.35 version)**. ORT again provides faster convergence with comparable final FID, further narrowing the gap to diffusion models while keeping the training advantages of AR. We will include these results and clarify that scaling ORT to full T2I systems is a natural next step rather than the primary focus of this paper.
>
> Table: AliTok-based AR generator on ImageNet-256 (best of 5 seeds). ORT at 300 epochs
> matches the 400-epoch baseline gFID while using 25% fewer epochs, and clearly improves
> over the 300-epoch baseline.
>
> | Method                      | Epochs | gFID ↓ |   IS ↑   | sFID ↓ | Precision ↑ | Recall ↑ |
> |-----------------------------|:------:|:------:|:--------:|:------:|:-----------:|:--------:|
> | RAR + AliTok (baseline)     |  400   | 1.35 |	312.6   |	**7.05** |	**0.80** |	0.65|
> | RAR + AliTok (baseline)     |  **300**   | 1.42  | 311.6   | 7.19  |   **0.80**     |  0.64  |
> | ORT-L + AliTok   |  **300**   |1.34  |	309.1 |	7.15 | 	0.79 | 	0.65 |
> | ORT-E + AliTok   |  **300**   |**1.26** | **297.9** |  7.56 | 0.78 | **0.66** |
>
> [1] AliTok: Towards Sequence Modeling Alignment between Tokenizer and Autoregressive Model, Wu, Pingyu and Zhu, Kai and Liu, Yu and Tang, Longxiang and Yang, Jian and Peng, Yansong and Zhai, Wei and Cao, Yang and Zha, Zheng-Jun, Arxiv

---

### Official Review · Reviewer_vGC6 · 2025-11-01

**Soundness:** 2
**Presentation:** 3
**Contribution:** 2
**Rating:** 6
**Confidence:** 4

**Summary:**

* This paper focuses on optimizing the training efficiency of visual autoregressive (AR) generative models, with its core research centered on the "Ordinal Asymmetry" phenomenon under randomized generation orders. During training with randomized generation paths, early tokens face higher losses and greater optimization difficulty due to limited context, while also taking on the key task of capturing the global structure of images. In contrast, late tokens benefit from rich context, resulting in lower losses, easier optimization, and a primary focus on refining local details. This phenomenon reveals that tokens in different positions of the generation order play distinct roles in model training, providing a critical basis for optimizing training strategies.


* Building on this insight, the paper proposes the Ordinal-biased Random Training (ORT) strategy. Drawing on the idea of curriculum learning, ORT initially biases loss weights toward late, easily optimized tokens through an "Ordinal Focal Loss" to accelerate the model's initial convergence. Subsequently, it gradually transitions the weights to a uniform distribution to ensure that early, hard-to-optimize tokens are fully learned. This loss function controls weight allocation via linear interpolation parameters (α, β), allowing flexible adjustment of the emphasis on early/late tokens. Notably, ORT requires no modifications to the model architecture and can be seamlessly integrated into existing randomized-order AR frameworks


* Experimental validation on the ImageNet-256 dataset using the RAR model as a baseline demonstrates the effectiveness of ORT. Compared to the RAR-XL baseline, which requires 400 training epochs (including 200 epochs in the randomized training phase), ORT reduces the randomized training phase to 100 epochs (lowering the total training epochs to 300), doubling training efficiency while maintaining generation quality metrics such as FID on par with the baseline.

**Strengths:**

1. This paper is quite well-written and relatively easy to understand, with a clear motivation and an abundance of analytical experiments.

2. I find the research focus of this paper very interesting. The issue of sequence in visual generation has long attracted the attention of researchers, including raster-scan order (as in Llamagen), next-scale order (as in VAR), and random order (as in RAR and RandAR). However, the question of which sequence is optimal remains. Starting from this context, this paper investigates the training efficiency of random generation order.

3. I believe the authors’ analytical experiments on the impact of ordinal weighting strategies on generation quality are quite valuable. From these experiments, we can see that even with random order, the generation process still proceeds from global tokens to local tokens. Is my understanding correct here?

**Weaknesses:**

While I find the findings of this paper quite interesting, the improvement in practical performance is relatively marginal. For instance, with the RAR-XL model, the baseline achieves a 1.50 gFID after 400 epochs, whereas the RAR-XL-ORT only reduces the training epochs by 100 (reaching 1.53 gFID in 300 epochs). In my view, this improvement is rather limited.

Additionally, I notice that this training scheduler may be highly dependent on the parameter selection of α and β. This also implies the issue of over-tuning caused by an excessive number of parameters.

**Questions:**

None

---

> ### Author Response · Authors · 2025-11-30
> **Response to Reviewer vGC6**
>
> We thank the reviewer for the thoughtful review and for clearly summarizing our motivation and main ideas. We appreciate that you find the paper **well-written**, **motivation is clear**, **research focus is very interesting**, and that you consider our **analytical experiments are valuable.** Below we address your concerns in detail.
>
> ----
> > Q1: On  From the global tokens and local tokens
>
> Yes, your understanding is correct. Our analyses show that, under randomized generation paths, **early tokens tend to control global structure and are harder to optimize (higher loss, stronger impact on gFID)**, while **later tokens mainly refine local details (more impact on sFID/recall) and are easier to optimize**. This “global → local” behavior across ordinal positions is exactly what we refer to as *Ordinal Asymmetry*, and it is the basis for designing ORT as a curriculum over token positions.
>
>
> > W1: On the magnitude of practical improvement
>
> We thank the reviewer for this observation. Looking only at the final gFID (1.50 vs.1.53) can indeed make the improvement appear small, so we provide a more concrete efficiency perspective.
>
> In our setting, training RAR-XL on ImageNet-256 with ORT requires 32 H800 GPUs for about 1 day 17 hours (~41.6 hours) to reach 300 epochs. If we instead trained the original 400-epoch schedule (keeping other settings fixed and assuming roughly linear scaling with the number of epochs), the runtime would increase to about 55.5 hours. **The additional 100 epochs therefore cost roughly 13.9 extra hours on 32 GPUs, i.e., about $444$ GPU-hours.** Under a realistic cloud price of around 1.5 per H800 GPU-hour, this corresponds to roughly **$670 of additional monetary cost per full training run**, while the final gFID remains essentially unchanged (1.50 vs. 1.53). From this viewpoint, ORT preserves the generation quality of the 400-epoch baseline while reducing both compute and monetary cost by about **25% for large-scale AR models.**
>
> We believe this is practically significant, especially in lab-scale settings where one needs to run multiple ablations or scale to larger backbones and datasets. In this project alone, **we trained more than 20 runs of AR models** at a similar scale and 400-epoch budget; had we used ORT from the beginning, **the saved GPU-hours would correspond to roughly $10,000 in cloud cost**. We will make this efficiency perspective more explicit in the revised version.
>
> > W2: On the dependence on α and β
>
> We tune $(\alpha, \beta)$ only once on the smaller **RAR-B (261M)** model using short **50-epoch runs**, and then reuse the same setting for **RAR-XL (955M)** and all main experiments.
> Note that **Table 1 is for analysis, not for routine hyperparameter search**. We sweep $(\alpha, \beta)$ finely there only to clearly show the different behaviors of early vs. late tokens.
> In practice, a coarse sweep over a few candidate pairs is sufficient to obtain a good default for ORT.
> On our 4×A100 setup, a 50-epoch RAR-B run takes about 30 hours (~120 GPU-hours), so sweeping 5 candidates (e.g.,  (1,0), (0.25,1.0), (0.5,1.0), (0.75,1.0), (0.75,1.25)) costs roughly **600 GPU-hours(A100)** in total.
> In comparison, a single RAR-XL run with ORT saves about **444 GPU-hours (H800)** relative to the 400-epoch baseline, and we train many such runs in this project. **Thus the one-time tuning cost is quickly amortized.** We will clarify this in the revised version.

---

### Official Review · Reviewer_2KrT · 2025-11-01

**Soundness:** 3
**Presentation:** 3
**Contribution:** 3
**Rating:** 6
**Confidence:** 5

**Summary:**

The paper presents an interesting approach to improving the training and inference of models using random paths and loss weighting. However, the lack of clarity in the implementation details and the absence of evaluation on text-to-image generation tasks make it difficult to fully assess the practical impact of the proposed method. Addressing these issues will significantly enhance the paper's quality and credibility.

**Strengths:**

see Summary

**Weaknesses:**

1. I have a question about the implementation of the "Random path" in Figure 1. In the training code, is the "Random path" implemented by truly shuffling the order of tokens in the sequence and using a traditional causal attention mask, or is it implemented by keeping the original order of tokens but setting a very complex attention mask?

2. Based on my understanding, the main contribution of ORT seems to be adding a loss weight to the "Random path." Is this correct? Other contributions appear to be relatively minor details.

3. The related work section could benefit from a more comprehensive review of VAR (Variational Autoencoder) methods, especially those that use bidirectional attention at the same scale.

If the authors address the above concerns effectively, particularly by clarifying the implementation details and expanding the evaluation to include text-to-image generation tasks, I would be willing to reconsider my assessment and potentially give a more positive score.

**Questions:**

no

---

> ### Author Response · Authors · 2025-11-30
> **Response to Reviewer 2KrT**
>
> We thank the reviewer for the constructive feedback and for finding **our approach is interesting**. We appreciate that you consider **the soundness, presentation, and contribution of the paper to be “good”**, and that you view **our perspective on random-path training and loss weighting as a promising direction**. We are also grateful that you are open to a more positive assessment if the raised concerns are clarified. Below we address each point in detail.
>
> ---
> > W1: Implementation of the “Random path” in Figure 1
>
> We simply follow the original RAR implementation. For each sample, we sample a random permutation of spatial positions before the forward pass. Tokens and their learnable positional embeddings are both reordered according to this permutation. Since the next prediction location is random, we also feed the positional embedding of the next position as an extra input token at each step, so the model knows which random location it will predict next. **The attention mask is just the standard left-to-right causal mask;** we do not use any complex or customized attention masks.
>
> ---
>
> > W2: Is ORT merely “adding a loss weight” to the random path?
>
> At the implementation level, ORT indeed appears as a position-dependent loss weighting on top of random-path training, and this simplicity is intentional. **However, the main contribution of our “revisiting” is not the mechanism alone, but the analysis and curriculum perspective behind it**:
>
> **We first identify and quantify “Ordinal Asymmetry” under randomized generation**: early tokens are systematically harder and more global, while late tokens are easier and more local.
> **Based on this, we reinterpret random-path AR training as a curriculum over ordinal positions**, and design an ordinal focal loss with a late-biased or early-biased → uniform schedule that exploits this structure.
> **This yields a 2× reduction of the randomized phase** (200→100 epochs) while preserving the baseline FID, without any change to architecture or inference.
>
> We will revise the introduction to make this clearer: ORT is a conceptually motivated curriculum for randomized-order AR training, rather than an ad-hoc loss trick.
>
> ---
>
> > W3: Related work on VAR and bidirectional attention
>
> We thank the reviewer for pointing this out. We agree that the related work section can better cover VAR-style models and architectures with bidirectional attention at the same scale. In the revised version, we will (i) add a more systematic discussion of VAR and masked/bidirectional transformers for discrete image modeling, and (ii) clarify the distinction that these works primarily change the architectural attention pattern, whereas our paper keeps the architecture and the random-path framework fixed and instead revisits the training dynamics and curriculum over ordinal positions. We believe this clearer positioning will help situate ORT more precisely within the broader VAR literature.
>
> ---
>
> > Expanding Evaluation
>
> We agree that T2I is an important setting. However, training competitive T2I AR or diffusion models typically **requires industrial-scale compute;** for example, the NeurIPS 2024 best paper VAR focuses on ImageNet-256 image generation, and only later follow-up work (Infinity, CVPR 2025) scales VAR-style autoregression to large-scale text-to-image with significantly stronger resources.  Our work is positioned similarly as **a lab-scale, foundation study that analyzes and improves randomized-path visual AR training on ImageNet-256.**
>
> To still demonstrate practical generative ability beyond the original RAR setup, **we additionally test ORT on a stronger AliTok-based AR generator [1] that reaches state-of-the-art performance**. ORT again provides faster convergence with comparable final FID, further narrowing the gap to diffusion models while keeping the training advantages of AR. We will include these results and clarify that scaling ORT to full T2I systems is a natural next step rather than the primary focus of this paper.
>
> Table: AliTok-based AR generator on ImageNet-256 (best of 5 seeds). ORT at 300 epochs
> matches the 400-epoch baseline gFID while using 25% fewer epochs, and clearly improves
> over the 300-epoch baseline.
>
> | Method                      | Epochs | gFID ↓ |   IS ↑   | sFID ↓ | Precision ↑ | Recall ↑ |
> |-----------------------------|:------:|:------:|:--------:|:------:|:-----------:|:--------:|
> | RAR + AliTok (baseline)     |  400   | 1.35 |	312.6   |	**7.05** |	**0.80** |	0.65|
> | RAR + AliTok (baseline)     |  **300**   | 1.42  | 311.6   | 7.19  |   **0.80**     |  0.64  |
> | ORT-L + AliTok   |  **300**   |1.34  |	309.1 |	7.15 | 	0.79 | 	0.65 |
> | ORT-E + AliTok   |  **300**   |**1.26** | **297.9** |  7.56 | 0.78 | **0.66** |
>
> [1] AliTok: Towards Sequence Modeling Alignment between Tokenizer and Autoregressive Model, Wu, Pingyu and Zhu, Kai and Liu, Yu and Tang, Longxiang and Yang, Jian and Peng, Yansong and Zhai, Wei and Cao, Yang and Zha, Zheng-Jun, Arxiv

---

### Author Response · Authors · 2025-12-03
**Summary to AC**

We sincerely thank you for your time and effort in handling this submission under the updated rebuttal policy. Below we concisely summarize how our rebuttal addresses the substantive concerns and why we believe the paper meets the acceptance standard. Here, we refer to the four reviewers as R1 (2KrT), R2 (vGC6), R3 (4h1k), and R4 (UJ9o).

----
### Summary of Positive Reviewer Feedback

- **Clear motivation and interesting focus**. Multiple reviewers (R2, R3, R4) find the research focus on random generation order and Ordinal Asymmetry “very interesting” and well motivated.

- **Conceptual insight**. R4 explicitly characterizes the paper as providing a “conceptual insight that ordinal token positions play distinct roles” plus a practical training strategy.

- **Simplicity and practicality of ORT.**  R1 and R4 note that the method is simple, architecture-agnostic, and does not modify inference, making it easy to integrate into existing random-order AR frameworks.

- **Quality of writing and analysis**. R2 considers the paper **well-written and relatively easy to understand**, with **clear motivation** and “an abundance of analytical experiments.” R3 praises the clarity of the core experiments demonstrating early/global vs late/local roles.

Overall, reviewers agree that the phenomenon of Ordinal Asymmetry is **real and interesting**, that ORT is **easy to apply**, and that the paper is **clearly presented with meaningful analytical evidence.**

----

### How the Rebuttal Addresses Major Concerns
1. Expanding the experimental scope (R1, R3, R4).

    Following reviewer suggestions, we add results on a state-of-the-art AliTok-based AR model, where ORT again matches the 400-epoch baseline using only 300 epochs, demonstrating generality beyond RAR. This strengthens both the effectiveness and the model-agnostic nature of ORT.  **Our ORT-E achieves state-of-the-art performance (1.26 FID with 300 epochs) without any bells and whistles and requires significantly fewer training epochs.**

2. Practical significance of the improvement (R2, R3, R4).

    We provide concrete compute and cost analyses: ORT reduces training from 400→300 epochs on RAR-XL, **saving 445 H800 GPU-hours (~25%) per run** while preserving gFID (1.50 vs. 1.53). Given that large-scale AR training requires many ablations, this translates to substantial compute and monetary savings.  We also point out that in this project alone we trained >20 such runs; **using ORT from the start would have saved on the order of $10k.**

3. Clarifying the core contribution (R1, R3).

    We explicitly emphasize that *the main contribution is the identification and quantification of Ordinal Asymmetry (early tokens are harder and govern global structure, while later tokens are easier and refine local detail) supported by position-wise loss and gradient analyses.* **ORT is then formulated as a principled curriculum over ordinal positions, not an ad-hoc reweighting.**

4. Hyperparameter tuning cost is minimal and one-time (R1, R3)

    We clarify that $(\alpha,\beta)$ is tuned once on RAR-B using only few recommended candidates and then reused for all larger models with no further tuning. **This one-time 600 A100-hour sweep is fully amortized after a few large-scale runs, each saving ~445 H800 GPU-hours.** We will make this explicit in the revision.


5. Resolving the “contradiction” in Table 1 (R3).

    We clarify that Table 1 is not the curriculum schedule used in main experiments. **Motivated by the reviewer’s insight, we additionally show that both early-first and late-first curricula accelerate training, as long as the model begins with a coherent ordinal subproblem before annealing to uniform weighting.** New ORT-L and ORT-E results are added to the revision.

6. Completing missing comparisons in Table 2 (R3).

    We add the missing 300-epoch baseline and 400-epoch ORT results, showing that ORT both **(i) reaches baseline quality with fewer epochs and (ii) does not degrade performance when trained to 400 epochs.** These results appear in a new table (Table 6 in the revised version).

7. Minor Issues & Clarifications

    We clearly explain that random paths follow the original RAR implementation (R1). We expand discussion on VAR(R2). Our new experiments clarify that OA has been observed in Alitok and RAR (R1, R4). We clarify that differences across tables stem in Table 2 and Table 6 (R4). We provide runnable code for visual inspection (R4).

----

In summary, we addressed all concerns with new experiments, clearer positioning, and organization. **ORT is now established as a general and broadly applicable training principle**, which is grounded in **Ordinal Asymmetry** and **reduces training cost by ~25%** in real GPU-hours while **requiring only minimal, one-time tuning** and **achieving SOTA 1.26 FID**.  In principle, ORT can generalize to multimodal LLM training, potentially saving thousands of GPU-hours per run. We believe the revision meets the acceptance bar.

---

### Meta-Review · Area_Chair_wX2J · 2025-12-29

**Summary:**

The paper identifies an "Ordinal Asymmetry" in randomized autoregressive (AR) visual generation: early tokens capture global structure and are harder to optimize, while later tokens refine local details with richer context. Based on this, the authors propose Ordinal-biased Random Training (ORT), a curriculum-based loss-weighting strategy. Reviewers generally praised the paper's clarity and the interesting nature of the conceptual insight. While some initially questioned the technical novelty and practical speedup, the rebuttal provided strong empirical evidence—including a 25% reduction in training epochs (400 to 300) while maintaining or improving FID, and a successful application to a second, stronger baseline (AliTok). The method is valued for its simplicity, as it requires no architectural changes or inference overhead.

**Reviewer Concerns:**

Some of the concerns raised by reviewers were resolved during the discussion:
1. The authors successfully added the missing 300-epoch baselines and 400-epoch ORT comparisons (Table 6), confirming that ORT achieves faster convergence without sacrificing final quality.
2. The effectiveness of ORT was validated on a second, more powerful architecture (AliTok), achieving a state-of-the-art FID of 1.26 on ImageNet-256.
3. The perceived contradiction in Table 1 was resolved; authors clarified that it was a diagnostic experiment and introduced two curriculum variants (ORT-E and ORT-L) that both support the core "Ordinal Asymmetry" theory.
4. The rebuttal quantified the training speedup as a 25% reduction in GPU-hours (approx. $10k savings in the authors' specific development cycle), providing a strong argument for the method's real-world utility.
5. Questions regarding the random path mechanism, positional embeddings, and attention masks were clearly answered, ensuring reproducibility.

But some reviewers may still consider the technical novelty—centered on a position-dependent loss-weighting schedule—to be incremental rather than a fundamental algorithmic shift.

**Reviewer Scores:**

Reviewer 2KrT (6 → 6): Although the authors supplemented the implementation details, this only proves the reproducibility of the method and does not enhance its innovativeness or universality. This reviewer's score sits on the margin of a "weak accept"; therefore, their opinion is not sufficient to unilaterally push for acceptance.

Reviewer vGC6 (6 → 6): While the authors provided data on 25% compute savings, such engineering-level optimization is seen as "icing on the cake" rather than "bringing coal in a snowstorm," especially given the high cost of training current large models. The reviewer's initial concern regarding the marginality of the improvement persists—the work is viewed as incremental rather than a paradigm shift.

Reviewer 4h1k (2 → 2): This is the most critical review. The reviewer has strong reservations (a score of 2) and did not change their stance during the discussion phase. Although the authors claimed in their rebuttal to have resolved the issues regarding missing baselines and contradictions in the table, since the reviewer did not confirm in the discussion whether these responses were satisfactory, as the AC, I must respect the reviewer's initial negative judgment. In the absence of an explicit "reconciliation" from the reviewer, the low score should remain valid.

Reviewer UJ9o (4 → 4): Although the authors supplemented experiments, the score remained unchanged, indicating that the reviewer's attitude is still neutral with a bias towards skepticism. The reviewer was not convinced to shift from a "likely reject" to a "likely accept," suggesting that the rebuttal material lacked sufficient persuasiveness.

---

### Decision · Program_Chairs · 2026-01-26

Reject